# Adversarial Data Generation of Multi-category Marked Temporal Point Processes with Sparse, Incomplete, and Small Training Samples

## Abstract

Asynchronous stochastic discrete event based processes are commonplace in application domains such as social science, homeland security, and health informatics. Modeling complex interactions of such event data via marked temporal point processes (MTPPs) provides the ability of detection and prediction of specific interests or profiles. We present a novel multi-category MTPP generation technique for applications where training datasets are inherently sparse, incomplete, and small. The proposed adversarial architecture augments adversarial autoencoder (AAE) with feature mapping techniques, which includes a transformation between the categories and timestamps of marked points and the percentile distribution of the particular category. The transformation of training data to the distribution facilitates the accurate capture of underlying process characteristics despite the sparseness and incompleteness of data. The proposed method is validated using several benchmark datasets. The similarity between actual and generated MTPPs is evaluated and compared with a Markov process based baseline. Results demonstrate the effectiveness and robustness of the proposed technique.

## 1 Introduction

Marked Temporal Point Processes (MTPPs) are widely used for modeling and analysis of asynchronous stochastic discrete events in continuous time (Upadhyay et al., 2018; Türkmen et al., 2019; Yan, 2019) with applications in numerous domains such as homeland security, cybersecurity, consumer analytics, health care analytics, and social science. An MTPP models stochastic discrete events as marked points $(e_i)$ defined by its time of the occurrence $t_i$ and its category $c_i$. Usually, point processes are characterized using the *conditional intensity function*, $\lambda^*(t) = \lambda(t|\mathcal{H}_t) = \mathbb{P}[event \in [t, t + dt]|\mathcal{H}_t]$, which given the past $\mathcal{H}_t = \{e_i = (z_i, t_i)|t_i < t\}$ specifies the probability of an event occurring at future time points. There are many popular intensity functional forms. Hawkes process (self-exciting process) (Hawkes, 1971) is a point process used in both statistical and machine learning contexts where the intensity is a linear function of past events ($\mathcal{H}_t$) (Türkmen et al., 2019). In traditional parametric models, the conditional intensity functions are manually pre-specified (Yan, 2019). Recently, various neural network models (generally called neural TPP) have been used to learn arbitrary and unknown distributions while eliminating the manual intensity function selection. Reinforcement learning (Zhu et al., 2019; Li et al., 2018), recurrent Neural Networks (RNN) (Du et al., 2016), and generative neural networks (Xiao et al., 2018) are used to approximate the intensity functions and learn complex MTPP distributions using larger datasets.

Recent advances in data collection techniques allow collecting complex event data which form heterogeneous MTTPs where a marked point $(e_{ij})$ defines a time of occurrence $(t_i)$ and a category $(c_j)$ separately. Therefore, multi-category MTTPs not only concern about the time of occurrence but also the category of the next marked point. The multi-category MTTPs append extra dimensionality to the distribution which complicates the learning using existing technologies. In fact, multi-category MTPPs are greatly helpful to model the behavioral patterns of suspicious or specific individuals and groups in homeland security (Campedelli et al., 2019b;a; Hung et al., 2018; 2019), potential malicious network activities in cybersecurity (Peng et al., 2017), recommendation systems in consumer analytics

(Vassøy et al., 2019), and the behavioral patterns of patients to determine certain illnesses (Islam et al., 2017; Mancini & Paganoni, 2019).

A number of challenges limit the collection and access to data in many fields often resulting in small and incomplete datasets. Scenarios involving social, political and crime behaviors are often incomplete due to data collection challenges such as data quality maintenance, privacy and confidentiality issues (National Institutes of Health & Services, 2020), but still a rigorous analysis with complete data is essential to produce accurate and reliable outcomes. So, there is a critical need for a technique to capture and learn from MTPP distribution, develop and apply machine learning algorithms, etc., for a small set of data some of which may be incomplete. We present an adversarial multi-category MTPP generation technique which is capable of generating sparse, asynchronous, stochastic, multi-category, discrete events in continuous time based on a limited dataset. Adversarial training has recently evolved and is able to provide exceptional results in many data generation applications, mostly in image, audio, and video generation while precisely mimicking the features of an actual dataset. The primary GAN architecture (Goodfellow et al., 2014) only engages well for continuous and complete data distributions and GANs have not been used for learning the distribution of discrete variables (Choi et al., 2017). Later, GAN architectures for discrete events have been introduced (Makhzani et al., 2015; Yu et al., 2017) and also applied for MTTP generation using extensive training data (Xiao et al., 2018; 2017).

Adversarial autoencoders (AAE) are fluent in capturing latent discrete or continuous distributions (Makhzani et al., 2015). In this work, we present feature mapping modules for accommodating incomplete data and make AAE capable of capturing the MTPP distributions of incomplete and small datasets. The incompleteness of the data points can be occurred in following ways. The marked points have been not collected or actors did not originally expose some marked points due to the dynamicity of these stochastic processes, which is the case especially in social and behavioral domains. Main contribution of the paper is a novel technique to synthetically generate high-fidelity multi-category MTPPs using adversarial autoencoders and feature mapping techniques by leveraging sparse, incomplete, and small datasets. To the best of our knowledge, there is no technique available for multi-category MTTP generation using such a dataset which is significantly more challenging than the existing generation scenarios.

Section 2 reviews related literature on MTTPs and AAEs. Section 3 presents the definition of multi-category MTTPs and Section 4 discusses the usage of AAEs for incomplete, multi-category MTTP generation. Then Section 5 presents the unique preprocessing and postprocessing techniques include in the feature mapping encoder and the decoder. Section 6 discusses the results of the experiment, and Section 7 summarises the conclusion and future work.

## 2 RELATED WORK

MTPPs are widely used for modeling of asynchronous stochastic discrete events in continuous time (Upadhyay et al., 2018; Du et al., 2016; Li et al., 2018; Türkmen et al., 2019). Usually, an MTTP is defined using a conditional intensity function (Türkmen et al., 2019) which provides the instantaneous rate of events given previous points. Intensity functions are often approximated by various processes such as the Poisson process, Hawkes process (self-exciting process) (Hawkes, 1971), and self-correcting process (Isham & Westcott, 1979). In traditional MTPPs, the intensity function has to be explicitly defined; however any mismatch between the manually defined and the underlying intensity function of a process can have a significant adverse impact on the accuracy of models and outcomes. Deep generative networks avoid the requirement of manually identifying the intensity and thus allows the use of arbitrary and complex distributions. Recurrent Neural Networks (RNNs) with reinforcement learning have been widely used in recent years (Du et al., 2016; Li et al., 2018) as well as several hybrid and extended models are also presented. A stochastic sequential model is proposed in (Sharma et al., 2019) as a combination of a deep state space model and deterministic RNN for modeling MTPPs. FastPoint (Türkmen et al., 2019) uses deep RNNs to capture complex temporal patterns and self-excitation dynamics within each mark are modeled using Hawkes processes. A semi-parametric generative model is introduced in (Zhu et al., 2019) for spatio-temporal event data by combining spatial statistical models with reinforcement learning. The advanced data collection techniques and online social media platforms produce complex event data and thus social network analysis can now be used to inform solutions to many societal issues (Bonchi et al., 2011). Many such

processes of heterogeneous and complex events require multi-category MTPP-based representation. The data integrity is also a major concern in social networks as many fake and misleading data is not uncommon (Muramudalige et al., 2019). In many disciplines, such as economics, biological and social sciences, removal of non-verifiable entries is crucial for maintaining the required data integrity, which in turn leads to incomplete and small datasets. Various techniques are introduced to handle missing data in different contexts (Folch-Fortuny et al., 2015; MacNeil Vroomen et al., 2016).

Generative adversarial networks (Goodfellow et al., 2014) have become an alternative for data generation without extensive problem specific theoretical foundation or empirical verification (Yan, 2019). The initial GAN architecture (Goodfellow et al., 2014) is capable of capturing the exact distribution of continuous and complete data but cannot be used for learning the distribution of discrete variables (Choi et al., 2017). The recent improvement in the form of Wasserstein GAN (Arjovsky et al., 2017) is used to implement generative TPP models (Xiao et al., 2018). *medGAN* is designed to learn the distribution of discrete features, such as diagnosis or medication codes, via a combination of an autoencoder and the adversarial framework (Choi et al., 2017). Adversarial Autoencoder (AAE) (Makhzani et al., 2015) is a probabilistic autoencoder which uses the GAN framework as a variational inference algorithm for both discrete and continuous latent variables. An aggregated posterior distribution of $q(z)$ on the latent code is defined with the encoding function $q(z|x)$ and the data distribution $p_d(x)$ as follows where $x$ denotes a input sample set.

$$q(z) = \int_x q(z|x)p_d(x)dx \tag{1}$$

In general usage of an AAE, $x$ represents consistent, discrete or continuous data samples where almost all data points are captured or completed in a given context. The challenge addressed in our paper is to apply an AAE for scattered and incomplete multi-category MTPPs generation using our proposed feature mapping techniques with a data approximation method. Details of such an AAE for sparse, incomplete, and multi-category MTPPs generation and feature mapping techniques are presented in Sections 4 and 5 respectively.

## 3    MULTI-CATEGORY MARKED TEMPORAL POINT PROCESSES

A marked temporal point process (MTPP) represents a certain set of asynchronous stochastic discrete actions/events in continuous time (Upadhyay et al., 2018; Li et al., 2018). Due to the immense availability of heterogeneous and complex event data in recent years, it is significant to model such complex events using multi-category MTPPs. A multi-category marked point is denoted as follows.

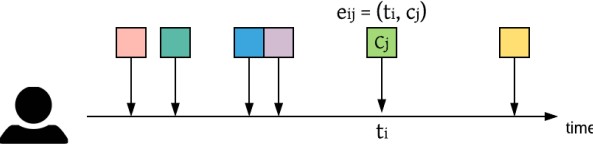

Figure 1: A basic representation of a multi-category MTPP, marked point colors depict multiple categories in a MTPP.

A marked point $e_{ij}$ is an **event** of category $c_j$ occurring at time $t_i$ as shown in Figure 1. Tables in Appendix A illustrate categories (i.e., $c_j$) in datasets, where these heterogeneous events exhibits complex dependencies and correlations. Usually, a MTPP analysis deals with an ensemble of marked temporal point processes (MTPPs). Multi-category MTPPs are described as follows. If the $k$th MTPP is $\mathcal{H}_k$ and its marked points are denoted as $e_{ij}^k \in \mathcal{H}_k$. Consider $n$ events (marked points) and $m$ categories in the $k$th MTPP, then marked points are characterized as $e_{ij}^k = (t_i, c_j)$ where $i \in [1, n]$ and $j \in [1, m]$. Then,

$$\mathcal{H} = \{e_{ij}^k = (t_i, c_j) \in \mathcal{H}_k; k \in [1, N]\}, \tag{2}$$

where $\mathcal{H}$ represents $N$ number of multi-category MTPPs. However, without loss of generality, we denote $\boldsymbol{e_{ij}^k}$ as $\boldsymbol{e_{ij}}$ in the following discussion.

In some problem domains, $i$ (time of occurrences) or $j$ (categories) values may change rapidly and some categories may not be recorded frequently. More importantly, with many problems in

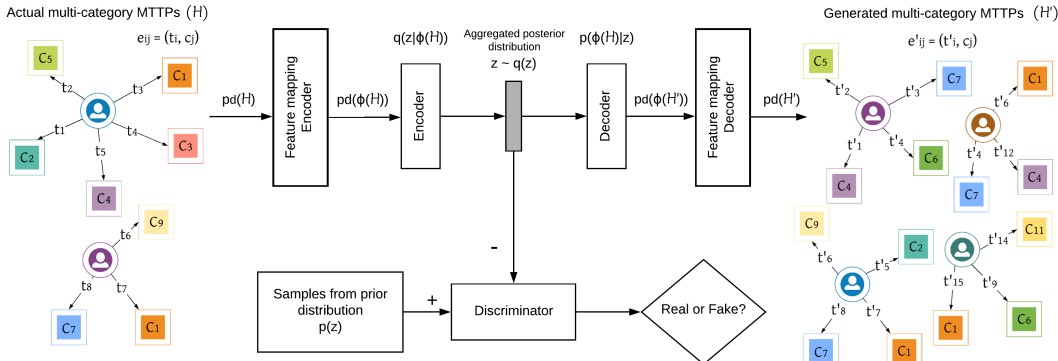

Figure 2: Architecture of an AAE for sparse, incomplete, and multi-category MTPPs.

domains such as social and behavioral sciences, not all marked points ($e_{ij}$) of a MTPP are known or observable due to the limitations of information gathering process, confidentiality constraints, unverifiability, deception, etc. A notable aspect of our work, is the use of sparse, incomplete, and multi-category MTPPs where an actor (an MTTP) either did not carry out activities corresponding to a certain event categories and marked points or they carried them out, but such activities were not reported in the reliable or permissible sources. To address these challenges, we propose a feature mapping encoder and decoder which are capable of capturing the sparseness and incompleteness of the data. The proposed feature mapping techniques consist of multiple steps including the calculation of cumulative probabilities for each category and a data approximation technique for incomplete data (briefly described in Algorithm 1). More details of the feature mapping encoder and the decoder are discussed in Section 5.

### 3.1 MULTI-CATEGORY MTTP DATASETS

We evaluate the performance of our technique using 3 real-world datasets from a diverse range of domains.

**Radicalization Dataset** The Western Jihadism Database (Klausen et al., 2020) has almost all the incidents of terrorist actions committed in western countries including timestamps and yet it does not consist of unconfirmed or undiscovered activities that cause an incomplete dataset. We use 135 detailed pathways (multi-category MTPPs) of home-grown jihadists (Klausen et al., 2018a) which have been extracted from radicalization trajectories of 335 known American jihadists (Klausen et al., 2016) as our real data distribution which covers over 24 behavioral indicators (categories). Table 1 in Appendix A describes the categories in the radicalization dataset.

**Mimic III Dataset** MIMIC III medical dataset (Johnson et al., 2016) is a large, freely-available database of clinical visit records of Intensive Care Unit (ICU) patients between 2001 and 2012. We only use the SERVICES data table, which describes the services that patients were admitted under. We extract 500 patient records as MTTPs across 6 different service types (categories). Service types are described in Table 2 in Appendix A.

**Stack Overflow Dataset** Stack Overflow is a question answering website, which attains badges to encourage user engagement and guide behaviors. We use publicly available archived dataset (Internet Archive, 2020) related to data science. Only 285 records (MTTPs) across 32 badges are extracted for our evaluations where the dataset is sparse, incomplete, and small. The details of the badges (categories) are shown in Table 3 (Appendix A).

## 4 ADVERSARIAL AUTOENCODER (AAE) FOR SPARSE, INCOMPLETE, AND MULTI-CATEGORY MTPPS

Figure 2 shows the adversarial autoencoder architecture for sparse, incomplete, and multi-category MTPPs. The actual and generated multi-category MTPPs are depicted in a tree structure where a root represents an actor of an MTTP. Square nodes stand for marked points, $c_j$ denotes the $j$th category,

and the value $t_i$ on an edge (connects an actor and a marked point) denotes the $i$th time of occurrence. The general AAE only includes an autoencoder and a discriminator (Makhzani et al., 2015), but our proposed architecture consists of an additional feature mapping encoder and a decoder to capture characteristics of the sparseness and incompleteness of the data. The *feature mapping encoder* transforms sparse, incomplete, and multi-category MTPPs ($\mathcal{H}$) to a cumulative distribution-based representation $\phi(\mathcal{H})$, which enables generating incomplete multi-category MTPPs through an AAE. The *feature mapping decoder* is able to rearrange the generated multi-category MTPPs $\phi(\mathcal{H}')$ to the actual format of multi-category MTPPs ($\mathcal{H}'$).

The autoencoder forces a compressed knowledge representation of the original input which reconstructs the same data distribution. Initially, the original data distribution of multi-category MTTPs $p_d(\mathcal{H})$ is fed into the *feature mapping encoder* which outputs the feature-mapped data distributions $p_d(\phi(\mathcal{H}))$. Then, the feature-mapped data is sent to the encoder where it compresses the data to a latent code vector $z$. $q(z|\phi(\mathcal{H}))$ and $p(\phi(\mathcal{H})|z)$ stand for the encoding and decoding distributions respectively. $q(z)$ represents the aggregated posterior distribution of hidden code which forms through the encoding function and the feature-mapped data distribution. An aggregated posterior distribution ($q(z)$) of the hidden code vector of an autoencoder for sparse, incomplete, and multi-category MTPPs can be defined as

$$q(z) = \int_{\phi(\mathcal{H})} q(z|\phi(\mathcal{H}))p_d(\phi(\mathcal{H}))d\phi(\mathcal{H}). \tag{3}$$

The operating principle of the AAE is that the autoencoder attempts to minimize the reconstruction error while the adversarial network tries to minimize the adversarial cost. Two simultaneous phases, *reconstruction phase* and *regularization phase* take place in each mini batch during training. The reconstruction phase relates to the autoencoder of the network, and it minimizes the data reconstruction error, often referred to as the loss. The regularization phase relates to the adversarial component of the network, where it minimizes the adversarial cost to fool the discriminator by maximally regularizing an aggregated posterior distribution $q(z)$ to the prior $p(z)$ distribution.

The simultaneous training process forces the discriminative adversarial network into thinking that the samples from hidden code $q(z)$ come from the prior distribution $p(z)$ (Makhzani et al., 2015). In these experiments, a normal distribution is used as the arbitrary prior $p(z)$. After the training process, the decoder defines a deep generative model that maps the prior distribution $p(z)$ to the feature-mapped data distribution $p_d(\phi(\mathcal{H}))$ and generates data samples $\phi(\mathcal{H}')$ from the prior and decoding distribution. The data generation can be interpreted as

$$p_d(\phi(\mathcal{H}')) = \int_z p(\phi(\mathcal{H})|z)p(z)dz, \quad \text{where } \phi(\mathcal{H}') \approx \phi(\mathcal{H}). \tag{4}$$

The generated feature mapped data ($\phi(\mathcal{H}')$) by the AAE is fed into the *feature mapping decoder* to transform the actual format of multi-category MTPPs ($\mathcal{H}'$). Further details of the feature mapping encoder and the decoder are discussed in Section 5.

## 5  FEATURE MAPPING

The major challenge of applying the AAE framework to multi-category MTPPs is that the data representation is structured; that is, each MTPP consists of a set of marked points belonging to various categories. In addition, each MTPP starts from an initial point (in the radicalization dataset, the date of birth of an actor) and continues with exposed multi-category marked points at different times. To implement the AAE framework discussed in the previous section, we propose a feature mapping which essentially maps the complicated MTPPs into the Euclidean space. This is achieved by a set of preprocessing steps including a data transformation for each category. The architecture of our proposed method is shown in Figure 2, and its functionality is summarized in Algorithm 1.

There are two major components: a feature mapping encoder (steps 1-3) and a decoder (steps 5-6) and the data generation (step 4). In **step 1**, all the marked point times ($t_i$) are transformed to the days ($a_i$) based on the initial point of each actor and shifted to a same days range. As a result, the updated marked points can be denoted as $e_{ij} = (a_i^{shifted}, c_j)$. For each MTPP, the value 0 is assigned to those categories that do not occur. As explained earlier, the absence of a certain category in an MTPP instance may be due to either that category not being associated with the MTPP, or

---

**Algorithm 1** Feature mapping & MTPP generation pipeline

---

**Input:** $\mathcal{H}$ (real MTTPs), $e_{ij} = (t_i, c_j) \in \mathcal{H}$
**Output:** $\mathcal{H}'$ (generated MTTPs), $e'_{ij} = (t'_i, c_j) \in \mathcal{H}'$
**Step 1** Convert timestamps to days $e_{ij} = (a_i^{shifted}, c_j)$
   **Step 2** Replace with percentiles $e_{ij} = (P_{ij}, c_j)$
     **Step 3** Data approximation technique for incomplete marks
       **Step 4** Data generation via AAE $p_d(\phi(\mathcal{H})) \to p_d(\phi(\mathcal{H}'))$ where $e'_{ij} = (P'_{ij}, c_j) \in \mathcal{H}'$
   **Step 5** Replace percentiles with actual values $e'_{ij} = (a_i^{'shifted}, c_j)$
**Step 6** Convert days to timestamps $e'_{ij} = (t'_i, c_j)$

---

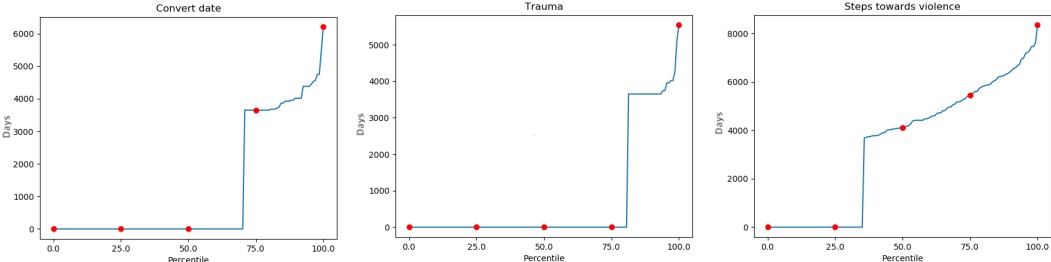

Figure 3: (Inverse) percentile graphs for categories in radicalization dataset ('Convert date', 'Trauma', and 'Step towards violence'). In this example, Y-axis represents days ($a_i$). Day 0 indicates the unavailability of marked points (eg: There are 70.37%, 80.74%, and 35.55% of unavailability of marked points for 'Convert date', 'Trauma', and 'Step towards violence' categories respectively).

limitations of data collection, or its presence not being recorded. The (inverse) percentile graphs for three categories (Convert date, Trauma, and Step towards violence) in the radicalization dataset are depicted in Figure 3. To measure the dissimilarity between two marked points in the same category, the difference of their shifted time is sufficient. However, such time difference varies a lot across different categories. In **step 2**, we propose to use the inverse percentiles instead. It helps to overcome the effect of sparseness and incompleteness of the marked processes to some extent. For instance, the shifted time of the marked point $e_{ij}$ can be transformed using $P_{ij} = F_{c_j}^{-1}(a_i^{shifted})$, where $F_{c_j}^{-1}$ is the inverse cumulative distribution function of the category $c_j$.

The inverse percentile distributions are shown in Figure 3 depict that there is a significance unavailability of marked points in categories. Therefore, unavailable marked points obtain higher percentile values which are sufficient to confuse the MTTP distribution. In **step 3**, we present a *data approximation technique* to further mitigate the effect of the sparseness and the incompleteness of the data by changing the percentile values only for unavailable marked points (where $a_i^{shifted} = 0$) using a uniform distribution. $P_{c_j}(0)$ denotes the percentile value of unavailable marked points in $j$th category. The percentile values ($P_{ij}$) of unavailable marked points are substituted by randomly generated values ($v_r$ where $v_r \in [0, P_{c_j}(0)]$) from the uniform distribution . Then, the percentile values are changed as follows. For a percentile value of $j$th category at $i$th time occurrence,

$$P_{ij} = \begin{cases} P_{ij} & \text{if } a_i^{shifted} \neq 0 \\ v_r \in [0, P_{c_j}(0)] & \text{otherwise} \end{cases} \quad (5)$$

After applying the *data approximation technique*, the updated marked points can be indicated as $e_{ij} = (P_{ij}, c_j)$. Steps 1-3 describe the sequential steps of pre-processing in the *feature mapping encoder* and produces feature mapped multi-category MTPPs $\phi(\mathcal{H})$ as the input to the AAE (see Figure 2). The feature mapped multi-category MTPPs can be denoted as follows. The $k$th feature mapped MTPP is $\phi(\mathcal{H}_k)$ and $e_{ij} \in \phi(\mathcal{H}_k)$, then

$$\phi(\mathcal{H}) = \{e_{ij} = (P_{ij}, c_j) \in \phi(\mathcal{H}_k); k \in [1, N]\} \quad (6)$$

where $\phi(\mathcal{H})$ is $N$ number of feature mapped multi-category MTPPs.

In **step 4**, feature mapped MTTPs $\phi(\mathcal{H})$ are fed to the AAE and data similar to the actual data is generated. The details of statistical methods that inspect the similarity between the datasets and the

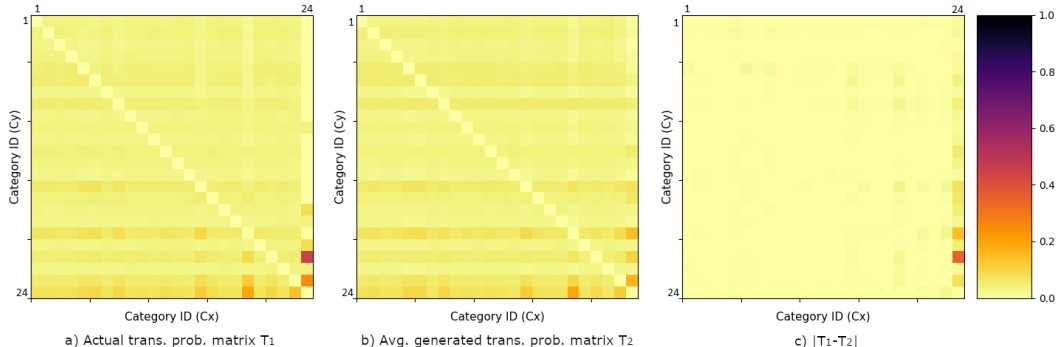

Figure 4: Conditional probability matrices for actual and generated datasets, and the difference. (radicalization data)

results are presented in the Section 6. Let $M$ denotes the number of MTTPs $\phi(\mathcal{H}')$, that are generated by the AAE. Then, the generated MTTPs can be denoted as follows.

$$\phi(\mathcal{H}') = \{e'_{ij} = (P'_{ij}, c_j) \in \phi(\mathcal{H}'_k); k \in [1, M]\} \tag{7}$$

Then, the *feature mapping decoder* (see Figure 2) is applied to transform generated percentile values to the actual multi-category MTPP format. In **step 5**, a generated percentile value of a marked point of $j$th category at $i$th time occurrence $P'_{ij}$ is converted to its day, based on the inverse percentile graphs in each category $F_{c_j}$ (Figure 3), and reshifted to the actual days range. Then, a marked point is redefined as, $e_{ij} = (a'_i, c_j)$. In **step 6**, pre-defined or any distribution-based on the initial point can be utilized to convert days to real timestamps $(t'_i)$ of the marked points. The advantage is that, it allows selecting any date range where provides flexibility to generate even future marked points based upon the requirement of the analysis. After the conversion, a generated marked point $(e'_{ij})$ of $j$th category at the $i$th occurrence $t'_i$ is denoted as $e'_{ij} = (t'_i, c_j)$. The $k$th generated MTTP is $\mathcal{H}'_k$ and $e'_{ij} \in \mathcal{H}'_k$, we can define the generated multi-category MTTP $(\mathcal{H}')$ as

$$\mathcal{H}' = \{e'_{ij} = (t'_i, c_j) \in \mathcal{H}'_k; k \in [1, M]\}. \tag{8}$$

The introduced feature mapping techniques generate multi-category MTPPs similar to the actual MTPPs. In Section 6, we show the similarity between AAE-based generated and actual MTPPs for radicalization using different statistical measurements.

## 6   RESULTS

In our experiment, the AAE is configured for 10K epochs with 32 mini-batch sizes in the training phase. The MTPP generation runs for 100 times and yields 10K MTPPs in each run. A `Keras Tensorflow` codebase[1] is used for the AAE implementation. The typical MTTP baselines like Reinforcement Learning, RNNs, Wasserstein GANs require a significant amount of data to train a network. Therefore, such techniques are not applicable to our proposed approach. As the baseline, we compare the proposed data generation technique with a Markov chain approach which was applied to the same dataset in (Klausen et al., 2018b). To compare with AAE-based generated MTTPs, we produce datasets using their conditional probability diagrams of the Markov chain by running 100 times and obtain 10K pathways in each run. Conditional probability calculation is performed after applying the data approximation technique (step 3) described in Section 5. Here, the pre-processed data $\phi(\mathcal{H})$ is used to calculate actual conditional probability which provides further validation on our proposed feature mapping techniques. In the same way, generated data $\phi(\mathcal{H}')$ by AAE (before enter the feature-mapping decoder) is utilized to calculate generated conditional probabilities.

Figure 4 depicts the conditional probability matrices for actual data (panel (a)), generated data (panel (b)), and their difference (panel (c)) for the radicalization data using a color map. In each matrix, a cell $(x, y)$ denotes the conditional probability $p(y|x)$ of category $y$ given category $x$. Here, the probability is calculated based on the marked points $e_{ij} = (t_i, c_j)$. The probability difference matrix

---

[1]https://github.com/eriklindernoren/Keras-GAN

highlights the accuracy and the robustness of our proposed generation method by showing that the real and generated conditional probabilities are almost the same for every pair of categories. The only noticeable difference arises in Category 24 (max val = 0.35), which is due to the fact that 'date of criminal action' is the last marked point for almost all observations. The marginal distributions for radicalization and mimic data in Figure 5 further corroborates the similarity of the actual and the AAE-based generated datasets. The generated marginals are calculated based on over 100 runs.

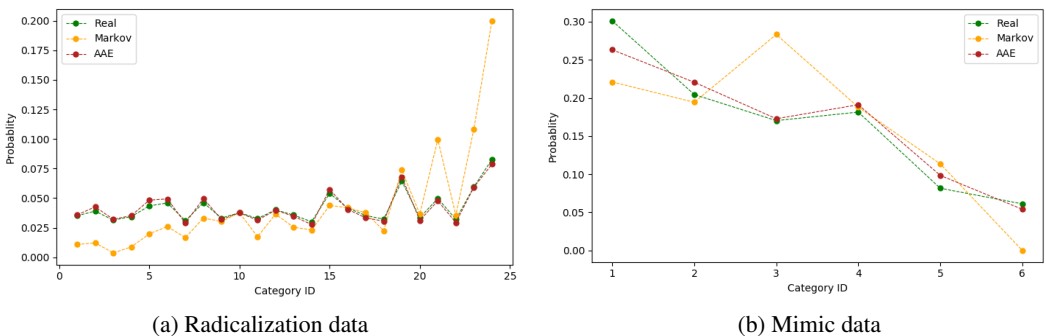

| (a) Radicalization data | (b) Mimic data |
|---|---|

Figure 5: Marginal distribution for each category: column sum for the conditional probability matrix; Real (green), Markov generated (orange), and AAE generated (red) data.

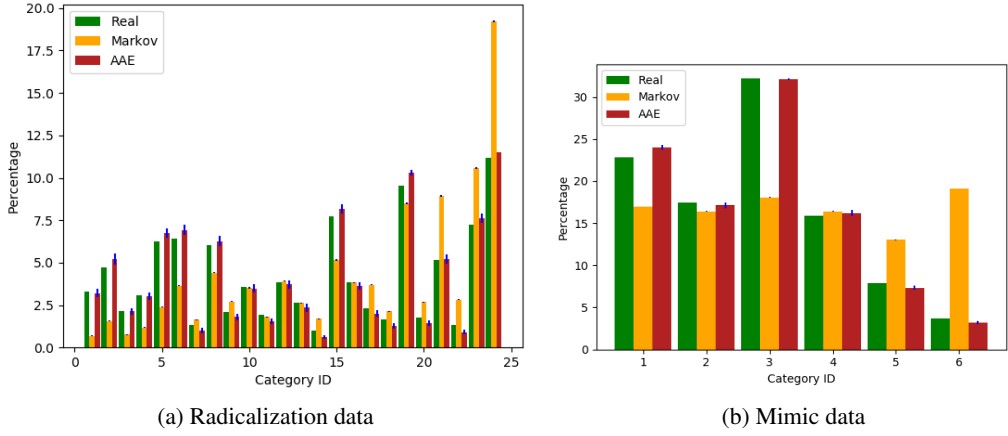

| (a) Radicalization data | (b) Mimic data |
|---|---|

Figure 6: Bar plots for the probability of occurrence for each category based on real (green), Markov (orange), and AAE (red) generated data. Error bars (blue) represent the standard error over 100 runs.

We further calculate the probabilities of occurrence for the categories described in Appendix A. Figure 6 compares the percentages of categories in actual and generated datasets. The error bar shows the standard error ($s.d./\sqrt{n}$ where $n = 100$) of each category and proves the robustness of the proposed multi-category MTPP generation technique. The 3rd (stackoverflow) dataset also performs the same in all experiments.

## 7    CONCLUSION & FUTURE WORK

We propose a novel multi-category MTPP generation technique using adversarial autoencoders for sparse, incomplete, and small datasets, where it is challenging to mimic an actual data distribution using existing techniques. The performance of our proposed method is demonstrated through real data examples. A cumulative distribution based preprocessing technique is introduced to capture the sequence pattern of the categories and reduce the dominance of the unavailable categories. The statistical similarity between the generated and actual data is demonstrated via diverse descriptive statistics. Ongoing work includes extending to the data anonymization applications and applying our proposed multi-category MTPP generation technique to address data privacy concerns.

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

# A    APPENDIX

The multi-category MTPP datasets

## 1. Radicalization dataset

| ID | Name | ID | Name | ID | Name | ID | Name |
|---|---|---|---|---|---|---|---|
| 1 | Convert Date | 7 | Rebellion | 13 | Dawa-Real Life | 19 | Desire for Action |
| 2 | Disillusionment | 8 | Lifestyle Changes | 14 | Epiphany | 20 | Passive Support |
| 3 | Trauma | 9 | Edu./Occ. Disen. | 15 | Peer-Immersion | 21 | Joins Foreign Org. |
| 4 | Personal Crisis | 10 | Drop-Out Date | 16 | Phy./Dom. Training | 22 | Issues Threats |
| 5 | Seeking Information | 11 | Underemployment | 17 | Marriage Seeking | 23 | Steps towards Violence |
| 6 | New Authority Figures | 12 | Dawa-Virtual | 18 | Societal Disengagement | 24 | Date of Criminal Action |

Table 1: Category IDs and category names in the radicalization dataset

## 2. Mimic III dataset

| ID | Name | Description |
|---|---|---|
| 1 | CMED | Cardiac Medical - for non-surgical cardiac related admissions |
| 2 | CSURG | Cardiac Surgery - for surgical cardiac admissions |
| 3 | MED | Medical - general service for internal medicine |
| 4 | SURG | Surgical - general surgical service not classified elsewhere |
| 5 | NSURG | Neurologic Surgical - surgical, relating to the brain |
| 6 | TRAUM | Trauma - injury or damage caused by physical harm from an external source |

Table 2: Category IDs and category names in the mimic III dataset

## 3. Stack Overflow dataset

| ID | Name | ID | Name | ID | Name | ID | Name |
|---|---|---|---|---|---|---|---|
| 1 | Nice Answer | 9 | Good Question | 17 | Student | 25 | Autobiographer |
| 2 | Enthusiast | 10 | Curious | 18 | Notable Question | 26 | Tumbleweed |
| 3 | Good Answer | 11 | Critic | 19 | Editor | 27 | Explainer |
| 4 | Excavator | 12 | Popular Question | 20 | Necromancer | 28 | Commentator |
| 5 | Nice Question | 13 | Yearling | 21 | Custodian | 29 | Promoter |
| 6 | Revival | 14 | Constituent | 22 | Caucus | 30 | Teacher |
| 7 | Quorum | 15 | Informed | 23 | Enlightened | 31 | Organizer |
| 8 | Famous Question | 16 | Scholar | 24 | Supporter | 32 | Patrol |

Table 3: Category IDs and category names in the stackoverflow dataset

