# OpenReview forum: "Adversarial Data Generation of Multi-category Marked Temporal Point Processes with Sparse, Incomplete, and Small Training Samples"
_ICLR.cc/2021/Conference — Reject_

### Official Review · AnonReviewer1 · 2020-10-20
**review for #895**

**Rating:** 3
**Confidence:** 4

**Review:**

This work studies the generation technique for multi-category MTPP using adversarial autoencoders for sparse and incomplete datasets. To address the sparsity and incompleteness, some pre-processing and post-processing methods are utilized to adapt the data to AAE.

Strengths: The model setup is reasonable, and the idea is straightforward to understand.

I recommend rejection of the paper for the reasons below.

Weakness: The major concern is the contribution of the work is very limited. The generation framework is a combination of adversarial auto-encoder and feature mapping encoder (decoder), where the AAE part is completely same as the original work in [Makhzani et al., 2015]. In my eyes, the only contribution is the addition of the feature encoder and decoder. Although the author named it like that, the so-called encoder and decoder are in fact some heuristic pre-processing and post-processing (shift and normalization) of the raw data.

Some specific concerns: the notation in Eq.(2) is confused. If c_j is defined as the category, then c_j should \in [1,m] not j. I understand what the author means by the current notation. It is better to define e_i=(t_i,c_i) where i \in [1,n] and c_i \in [1,m].
In algorithm 1, the steps 1-3 are some heuristic pre-processing of the raw data. Why do that? Why change the timestamps to days and scale it to a probability value? Any intuition behind this operation? My understanding is the raw data is shifted, scaled and normalized to [0,1]. Add some theoretical analysis why performing that pre-processing will help AAE handle the sparse and incomplete data.
In experiments, there is only one baseline model to compare with. Add more DNN-based generative baseline models will make the experiment more convincing.
In experiments, although the author claimed 3 real datasets, I did not see any experimental results of the 3rd (stackoverflow) dataset.

Typo: above Eq.(5), significance-->significant

---

> ### Author Response · Authors · 2020-11-17
> **Response to reviewer 1**
>
> Thanks for your valuable comments. The major contribution of the paper is a novel technique to synthetically generate high-fidelity multi-category MTPPs using adversarial autoencoders and feature mapping techniques by using sparse, incomplete, and small datasets. To the best of our knowledge, there is no technique available for multi-category MTTP generation using such a dataset which is significantly more challenging than the existing generation scenarios. In this paper, we use 3 real-world datasets and as an example, we will discuss the number of reasons to select the radicalization dataset. It is a data set that has been gathered and curated by a significant number of social scientists over the years and is being used by social scientists and low-enforcement. In spite of the significant manual effort (including reading court documents, texts, etc., and using specialized skills to mark up, etc., is going to be incomplete, sparse, and small due to covert nature of the domain, privacy issues, etc. This is exactly the type of problem that we want to address, rather than cases where plathora of data is available. It is these challenges that motivated us to develop this specific method proposed.
>
> We agree that the proposed feature-mapping encoder and decoder perform heuristic pre-processing and post-processing (shift and of the raw data to enable deep neural network to precisely capture the underlying distribution of spare and incomplete MTTPs using smaller amount of data. We denote them as feature-mapping encoder and decoder to formalize the data transforming process more conveniently.
>
> c_j is defined as the category and category is a name as depicted in Appendix 1.  Therefore, j \in [1,m] where m is the number of categories in a particular dataset. We think this sorts out the confusion about categories.
>
> As discussed in the paper, the steps 1-3 of pre-processing raw data is essential to mitigate the sparseness and the incompleteness of the data by mapping values to consistent data space. To generate a high-fidelity MTTP dataset, we must consider the order of occurrence of the marked points. Therefore, we have to embed that information into our dataset. By converting timestamps to days (based on a fixed reference) provides the time length of the marked point that enables to capture the order of occurrences. Figure 4 depicts the accuracy of the order of occurrence in generated MTTPs through conditional probability matrices and claim the importance of timestamp to days conversion.
>
> We will add some theoretical analysis of how the pre-processing will help AAE handle the sparse and incomplete data in the camera-ready version. As explained in the paper, this is the first proposed MTPP generation technique that leverages incomplete, sparse, and small data. Therefore, we cannot use DNN-based generative baseline models directly but we have to include our feature mapping technique. As the initial paper, we believe that is the out of the scope of the current paper and we will experiment with other DNN baselines with our feature mapping encoder and decoder in the next version of the paper.
> We mentioned that the 3rd (stackoverflow) dataset also performs the same in all experiments. We didn’t have enough space to add experiment results of 3rd dataset and we will add them in the camera-ready version where we have another additional page.  Thanks much for pointing out the typo and we will correct it before the discussion phase.

---

> > ### Comment · AnonReviewer1 · 2020-11-19
> > **Re: author response**
> >
> > Thanks for the authors' response. The authors' response partly confirmed my rating. I recommend the author to put all extra experimental details into the appendix, rather than stating "the 3rd dataset also performs the same in all experiments" which is not convincing. I still feel that the contribution of the paper is very limited for ICLR and would like to keep my original rating.

---

### Official Review · AnonReviewer4 · 2020-10-28
**The paper presents a deep method for a practical data augment for event sequences via adversarial autoencoder (AAE). The method is straightforward, which makes the contribution of the paper marginal. The paper also has some theoretical concerns.**

**Rating:** 5
**Confidence:** 4

**Review:**

The paper is concentrated at dealing with the data missing problem of MTPP and applies an AAE for the “incomplete multi-categorical MTPPs”.  First, the problem description appears questionable. Point processes are a class of stochastic processes for modelling discrete event sequences in a continuous time domain. They are statistical models and have a well-defined mathematical meaning. The expression of “incomplete point process” is quite confusing. One possible reason is that the authors fail to distinguish between the model and the data. One can say “incomplete data” or “incomplete observations of a model”, but “incomplete model” is not acceptable unless properly defined. Therefore, the proposed method seems not specifically for point processes but for the sequential data.

The authors seem to misunderstand the difference between empirical distribution and probabilistic distribution. The probabilities, e.g., $p_d(\mathcal{H})$, are actually empirical distributions of the time. Here the authors see the arrival time of the events as a random variable, and $t_{ij}$’s are independent samples of the random variable, which is unrelated to point processes. The percentiles used in the algorithm are essentially the cumulative empirical distributions of the arrival time. If point processes need to be considered here, the authors should define their probabilistic structures as they are probabilistic models, where the intensity function is often inevitable unless otherwise defined. I believe the confusing mathematical formulation should be considered as a fatal flaw.

---

> ### Author Response · Authors · 2020-11-17
> **Response to reviewer 4**
>
> Thanks for your valuable comments regarding point processes, based on further research we realized that these need to be further clarified. There are a different set of definitions to point processes.
> *A specific definition*
> “A temporal point process is a stochastic, or random, process composed of a time series of binary events that occur in continuous time (Daley and Vere-Jones, 2003)”
> *A general definition*
> “Point process models are useful for describing phenomena occurring at random locations and/or times.” http://www.stat.ucla.edu/~frederic/papers/encycpiece
> What we meant in the paper is that the general definition. Further, we discuss multi-category point processes that cannot be mapped to a single category point process directly. In the next version of the paper, we will further clarify this matter.
>
> We also believe that we didn’t confuse the reader about incompleteness. In the paper, we always discuss the incomplete data due to the various real-world challenges. We did not use the ‘incomplete model’ in the description, By incomplete data, what we mean is that all the categories of events are not available in point processes. In this work, we propose a novel technique to generate MTTPs using small, spare, and incomplete data. The idea is clear to all other reviewers too.
> p_d(H) are actually empirical distributions of the time and we denote p_d(H) as the data distribution to align with adversarial autoencoder notations in [Makhzani et al., 2015] to explain the adversarial process smoothly. Further, we use a non-parametric approach (generally called neural TPPs) using a deep adversarial network for MTTP generation where we don’t need to explicitly identify or define a probabilistic structure.

---

> > ### Comment · AnonReviewer4 · 2020-11-18
> > **Re: Authors' response**
> >
> > Thank you for the response, which addresses my concerns in part. Since $p_d (H)$ is still the empirical distributions of the interarrival time, the proposed method is essentially equivalent to a generic renewal process and the renewal intervals are iid, i.e., the $t_i$ you generated are not dependent on location or time. I would like to raise my rating a bit as the response cleared some of my concerns. However, I feel that the contribution of the paper is still not significant enough for acceptance.

---

### Official Review · AnonReviewer3 · 2020-10-31
**Justifications and experiments can be improved**

**Rating:** 5
**Confidence:** 4

**Review:**

Summary:
The authors propose a method for multi-category marked temporal point processes (MTPPs) generation with sparse, incomplete, and small training dataset. They apply Adversarial Autoencoder (AAE) and feature mapping techniques, which include a transformation between the categories and timestamps of marked points and the percentile distribution of the category.  The paper shows effectiveness and robustness of the proposed method by comparing with Markov chain approach on three datasets: Radicalization Dataset, Mimic III Dataset and Stack Overflow Dataset.

Strength:
1.	The paper is clear in general. Firstly, they define MTPP and multi-category MTPP. Then, the multi-category MTPP generation with sparse, incomplete, small training data problem is addressed.  The authors argue that it is a popular real-world problem where we only can access small and incomplete data.
2.	The description of the overall method is reasonable. But more details can improve understanding.

Weakness:
1.	The authors should provide justification of choosing to use AAE in the work. In particular, why is AAE an attractive approach for MTPP?
2.	 The authors mention: "The typical MTTP baselines like Reinforcement Learning, RNNs, Wasserstein GANs require a significant amount of data to train a network. Therefore, such techniques are not applicable to our proposed approach. As the baseline, we compare the proposed data generation technique with a Markov chain approach which was applied to the same dataset in (Klausen et al., 2018b)." The authors could try to apply Reinforcement Learning, RNNs, Wasserstein GANs with data filled by simple methods, to empirically validate that these methods are not suitable for incomplete, small data.
3.	In feature mapping method, the authors could provide justification for converting marked point times t_ij to days a_i. Besides, the method to convert t_ij to a_i and examples are not provided. If this is a common preprocessing method in MTPP, the authors should cite the relevant work. Overall, Step 1 of Algorithm 1 is not clear.
4.	At the data approximation technique (step 3 of the Algorithm 1), the author randomly chooses a probability for the appearance of an unobservable data point but there is a lack of explanations. Can the authors explain the reason of selecting from [0, Pcj(0)]?
5.	The authors should provide more details of incomplete and small dataset, and compare with their method when training with full data. This is to understand that if generated data is still good when training with small dataset


===============
after rebuttal: I thank authors for the responses. After reviewing the authors' response and other reviewers' comments, I keep my original rating.

---

> ### Author Response · Authors · 2020-11-17
> **Response to reviewer 3**
>
> In this work, we discuss MTTP datasets that are inherently incomplete, sparse, and small because of data collection and privacy issues. This work has a lot of potentials to be applied in broader applications such as social networks, phishing detection, network flow detection, etc. We have already applied the proposed technique for phishing detection, network flow detection and we were able to obtain promising results in particular domains. Developing a parametric model for incomplete, sparse, and small MTTPs is a harder task. Therefore, we proposed a non-parametric deep learning model for MTTP generation using proposed feature mapping techniques. The intuition behind our work is feature mapping techniques where it enables adversarial autoencoders (AAEs) to generate high-fidelity, incomplete, and sparse MTTPs. To the best of our knowledge, there is no other MTPP generation technique that leverages incomplete, sparse, and small data. The proposed feature mapping techniques are applicable for other deep learning data generation networks such as RNNs, Wasserstein GANs. Detailed comparison of such networks was not included here due to the page limitation. Details will be added to the camera-ready paper where we have an additional page. Further, the focus of this paper is to propose a novel generation model for incomplete, sparse, and small MTTPs.  In the extended version of this paper, we will try Reinforcement Learning, RNNs, and Wasserstein GANs with our feature mapping techniques.
>
> To generate a high-fidelity MTTP dataset, we must consider the order of occurrence of the marked points. Therefore, we have to embed that information into our dataset. By converting timestamps to days (based on a fixed reference) provides the time length of the marked point that enables to capture the order of occurrences. Figure 4 depicts the accuracy of the order of occurrence in generated MTTPs through conditional probability matrices and claim the importance of timestamp to days conversion.
>
> As shown in Figure 3, Pcj(0) represents the percentage of unavailable values in a particular category, which is a considerable value. If we feed percentiles to the AAE as it is, the values only lie between [Pcj(0), 100] and it is a small value range that leads the deep neural network is unable to capture the underlying distribution of marked points precisely. Therefore, we introduce a uniform random distribution for Pcj(0) values where the values uniformly distributed between [0,Pcj(0)]. In that way, the values are distributed well without changing the original distribution of the data that enables deep neural networks to generate high-fidelity percentile distribution.
> We provided more details of incomplete and small datasets in Appendix 1. As we discussed in the paper, we only use small datasets for training. The number of records in the training data are as follows.
>
> 1.	Radicalization dataset – 135 records
> 2.	Mimic III dataset – 500 records
> 3.	Stackoverflow dataset – 285 records
>
> We will add this and additional details to clarify points in reviewers’ comments 3,4 and 5, to the final version of the paper. We will use the additional page to accommodate these clarifications.

---

### Decision · Program_Chairs · 2021-01-11
**Final Decision**

**Decision:**

Reject

**Comment:**

This paper proposes a new generation technique for multi-category marked temporal point processes.  The paper was reviewed by three expert reviewers who expressed concerns for limited novel contributions, theoretical justification, and empirical evidence. The authors are encouraged to continue research, taking into consideration the detailed comments provided by the reviewers.